# Polyethylene glycol precipitation is an efficient method to obtain extracellular vesicle-depleted fetal bovine serum

Peng Wang[1], Onno J. Arntz[1], Johanna F. A. Husch[2], Van der Kraan P. M.[1], Jeroen J. J. P. van den Beucken[2], Fons A. J. van de Loo[1]*

1 Experimental Rheumatology, Radboud University Medical Center, Nijmegen, Netherlands, 2 Department of Dentistry Regenerative Biomaterials, Radboud University Medical Center, Nijmegen, Netherlands

* Fons.vandeloo@radboudumc.nl

**Data Availability Statement:** All relevant data are within the manuscript and its Supporting Information files.

## Abstract

Mesenchymal stromal/stem cell derived-extracellular vesicles (MSC-EVs) have gained interest as drug delivery nanoparticles, having immunoregulatory and potentiating tissue repair property. To maintain growth of MSCs and obtain pure MSC-derived EVs, the culture media should contain fetal bovine serum (FBS) devoid of EVs, as the presence of FBS EVs confounds the properties of MSC-EVs. Therefore, we tested three methods: 18h ultracentrifugation (UC) and ultrafiltration (UF), which are common FBS EV depletion methods in current MSC-EV research, and polyethylene glycol (PEG) precipitation to obtain three EV depleted FBS (EVdFBS) batches, and compared them to FBS and commercial (Com) EVdFBS on human adipose stem cell (hADSC) growth, differentiation, enrichment of EVs in hADSC supernatant and their biological function on collagen metabolism. Our comparative study showed UC and UF vary in terms of depletion efficiency and do not completely deplete EVs and affects the growth-promoting quality of FBS. Specifically, FBS EV depletion was comparable between PEG (95.6%) and UF (96.6%) but less by UC (82%), as compared to FBS. FBS protein loss was markedly different among PEG (47%), UF (87%), and UC (51%), implying the ratio of EV depletion over protein loss was PEG (2.03), UF (1.11), and UC (1.61). A significant decrease of TGFβ/Smad signaling, involving in MSC growth and physiology, was observed by UF. After 96 hours of exposure to 5% FBS or 5% four different EVdFBS cell growth media, the osteogenesis ability of hADSCs was not impaired but slightly lower mRNA expression level of Col2a observed in EVdFBS media during chondrogenesis. In consistent with low confluency of hADSCs observed by optical microscope, cell proliferation in response to 5% UF EVdFBS media was inhibited significantly. Importantly, more and purer ADSCs EVs were obtained from ADSCs cultured in 5% PEG EVdFBS media, and they retained bioactive as they upregulated the expression of Col1a1, TIMP1 of human knee synovial fibroblast. Taken together, this study showed that PEG precipitation is the most efficient method to obtain EV depleted FBS for growth of MSCs, and to obtain MSC EVs with minimal FBS EV contamination.

**Funding:** The authors received no specific funding for this work.

**Competing interests:** NO

## Introduction

Increasing attention has been paid on MSC-based therapy due to its anti-inflammatory, oxidative stress, apoptotic, and angiogenic effects [1, 2]. Specifically, the immunoregulatory and regenerative properties of MSC-sourced secretome have been persistently reported [3, 4]. Compared to parent cells which might cause immune responses and tumor formation [5, 6], extracellular vesicles (EVs) derived from MSC, one of secretome factors, are more effective and stable. Through transferring various contents such as nucleic acids, proteins, and lipids into recipient cells, MSC-EVs regulate phenotype, function, survival of resident cells and homing of immune cells, and participate in tissue maintenance and repair such as cartilage restoration and wound healing [7–9]. However, the main concern of enrichment of MSC-EVs from cell culture media is the contamination with exogenous EVs derived from fetal bovine serum (FBS). The presence of FBS-EVs may confound the therapeutic or diagnosis analyses of EVs derived from cultured cells [10, 11].

Based on an international survey conducted by International Society for Extracellular Vesicles (ISEV) in 83% of in vitro and preclinical EV studies, cell culture media was used to enrich various cell-EVs [12]. Furthermore, FBS is used as a common and important additive in MSCs culture, because it contributes to stable cell adhesion and alleviates cellular stresses associated with in vitro environment through providing various serum factors and other constitutes [13, 14]. Therefore, to date an increasing number of studies use EV depleted FBS (EVdFBS) to support growth of MSC, and obtain pure MSCs-EVs [7, 15, 16].

The common FBS-EVs depletion method used to obtain EVdFBS is performing ultracentrifugation (UC) at 120,000g for 18h in diluted FBS and considered to be the gold standard [17]. Shelke and colleagues conducted a study on EV depletion protocol using different centrifugation times for FBS and observed that 18 h of UC was the optimal time, even though it did not completely eliminate EV contaminants from FBS [18]. Pham *et al* later challenged the UC protocol and investigated the particles remaining in FBS after EV depletion through UC, showing that certain nanoparticles were not depleted after 18 h of UC [19]. For UF FBS EV depletion, Kornilov *et al* filtered FBS through ultra-15 centrifugal filters (55 minutes at $3000 \times g$) to obtain UF EV depleted FBS. Transmission electron microscopy revealed UF EV depleted FBS had no detectable EVs, distinguishing it from commercially EV-depleted FBS and UC EV depleted FBS. In addition, they observed UF EVdFBS contained no any protein, but a distinct protein pattern was detected in UF isolated FBS EVs in the study [20]. Of note, considering retrieve of a large scale of FBS volumes [21], methods used to deplete the EVs form FBS should be a single step procedure that can be used to process large volumes, and methods such as size exclusion chromatography (SEC isolation), or affinity-based isolation methods are either costly or not applicable. A drawback of using commercially available EV-depleted FBS is that the method is not made public. Given that certain studies have revealed contaminants in commercial EV-free FBS, caution should be exercised when using such material for in vitro EV studies [22].

Therefore, current techniques not only incompletely deplete EVs but also may remove essential FBS proteins. During the FBS-EVs depletion process, the serum factors (proteins, lipids, and lipoproteins) tend to become aggregated and removed from FBS, leaving a less potent and incomplete cell culture media supplement. Studying the best method for high selective FBS-EVs depletion is necessary. In this study, we tested three techniques: UC, UF, and polyethylene glycol (PEG) precipitation to obtain three different EVdFBS batches and compared them to FBS and commercial (Com) EVdFBS on human adipose stem cell (hADSC) growth, differentiation, enrichment of EVs in hADSC supernatant and their biological function on collagen metabolism.

## Materials and methods

### Preparation of different EV depleted FBS (Fig 1A)

**Ultracentrifugation method.** Culture media or PBS was used to dilute FBS before performing UC at 120,000g for 18h. UC EVdFBS was obtained from the supernatant (approximately 90%), and the precipitate on the bottom was denoted as UC isolated FBS EVs.

**Ultrafiltration method.** Similarly, after 55 min centrifugation at 3000g, FBS in the Amicon ultra-15 centrifugal filters (ref: UFC910024, 100kDa Merk Millipore Ltd.) had been separated in two parts: the permeate on the external tube ascribed as UF EVdFBS and UF isolated FBS EVs in the retentate room.

**PEG precipitation method.** To obtain PEG EVdFBS, FBS were supplemented with 50% w/v stock solution of PEG 6000 (Sigma-Aldrich, Taufkirchen, Germany) to the various final concentrations of PEG. The PEG EVdFBS could be collected from the supernatant of mixture

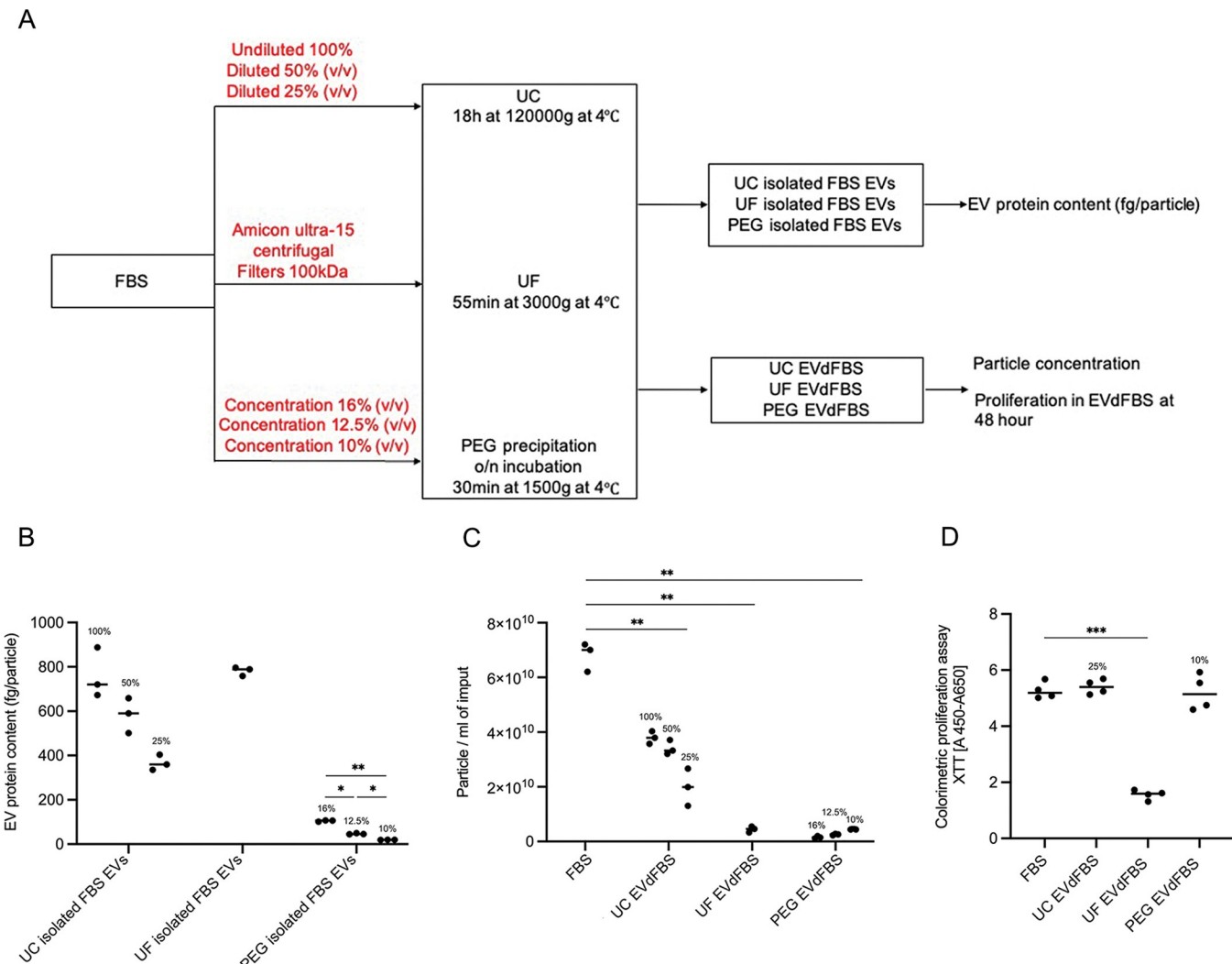

**Fig 1. Comparison among EV depletion methods.** (A) The working procedure to obtain UC/UF/PEG EVdFBS. (B) The EV protein contents of UC/UF/PEG isolated FBS EVs was calculated based on protein level determined by microBCA and particle concentration measured by NTA (in triplo). (C) Particle concentration of corresponding UC/UF/PEG EVdFBS measured by NTA (in triplo). (D) hADSCs proliferation at 48-hour in 5% FBS or UC/UF/PEG EVdFBS determined by XTT assay (in quadruple). For panel C, the p was determined by student two tailed t test, values are mean. And for panels B and D were calculated by one way ANOVA tests. ** $p < 0.01$.

after incubation overnight at 4°C and 30 min centrifugation at 1500g. Also, the pellet was regarded as PEG isolated FBS EVs on the bottom.

## Nanoparticle tracking analysis (NTA)

Particle size distribution was estimated by the Brownian motion of the particles in a NanoSight 300 using Nanoparticle Tracking Analysis 2.3 software (Nanosight Ltd, Amesbury, UK). Particles were diluted in PBS, till a suitable concentration for analysis was reached, and the concentration value between 20 to 80 particles/frame was deemed as effective measurement. Major particle concentration was evaluated for the particles between 30–300nm in diameter. 30s video was recorded using camera level 10 and detection threshold 5.

## Protein content

Total protein content of FBS and four different EVdFBS batches as well as three isolated FBS EVs was evaluated by the bicinchoninic acid micro-BCA protein assay kit (Pierce, Rockford, IL, USA). Using the 96-well plates, protein analysis was carried out according to the manufacturer's instructions, and the 2mg/ml BSA was the standard protein sample.

## XTT assay

To examine the quantification of human adipose tissues (hADSCs) proliferation cultured in FBS and four different EVdFBS batches, the Cell Proliferation Kit II (XTT) was used (Dojindo, US). The XTT reagent, which is a tetrazolium-based compound, is sensitive to cellular redox potential and cellular respiration converts this substrate into an orange-colored formazan product. The assay was done according to the manufacturer's instructions. After culturing hADSCs for 48 hours or 5 days, cellular metabolic activity was determined. By adding XTT label reagent, absorbance was measured after incubation for 4hr at 37*C at wavelength 450 nm using a microplate reader. The blank well on the same row and the wavelength >650 nm were taken as the references.

## Western blot

To detect the EV-specific markers Alix, CD81 and HSP-70 as well as the lipoprotein marker ApoE in the three isolated FBS EVs sample, equal amounts of protein were loaded onto 12% SDS-PAGE, together with protein ladder, and subsequently transferred onto a 0.45μm nitrocellulose membrane for 120 min at 350mA. The membrane was blocked with 5% skim milk for 1 h at room temperature. After washing 3 times with Tris-buffered saline containing 0.1% Tween 20 (TBST), the membranes were incubated, respectively, with 1:500 anti-Alix (Santa Cruz Biotechnology, Inc., Dallas, TX, USA; Cat # sc-53540, 1:2000 anti-CD81 (Santa Cruz Biotechnology, Inc., Dallas, TX, USA; Cat # sc-166029), 1:2000 anti-HSP70 antibody (Santa Cruz Biotechnology, Inc., Dallas, TX, USA; Cat # sc-32239) and 1:1000 anti-ApoE (Sanbio, Rosemont, USA; Cat # 18254-1-AP) at 4°C overnight. After washing in TBST for 3 times (5 min for each time), the membrane was incubated with horseradish peroxidase-linked secondary antibodies (anti-mouse or rabbit) for 2 h at room temperature. After washing five times with TBST, proteins were detected with ECL western blotting detection reagent (GE Healthcare, UK; Cat # lot 16961643) according to the manufacturer's instructions.

## CAGA12-Luciferase reporter assay [23]

To examine serum growth factor TGFb in FBS and four different EVdFBS batches, murine embryonic fibroblasts (3T3 fibroblasts) were seeded at 30.000 cells per 96-well plate and

transduced with SMAD-sensitive CAGA-Luc reporter adenovirus (kindly provided by Peter ten Dijke, Dept. Molecular Cell Biology, Leiden University Medical Center, Leiden, The Netherlands) the next day for 2.5 hours in serum-free DMEM condition. Then after a 20-hour recovery period using DMEM containing FBS, Cells were serum deprived for 8 hours and then were stimulated with DMEM alone (control) or with recombinant human TGF- β1 or FBS and four different EVdFBS batches, respectively. Using 50 µl reporter assay lysis buffer, cells were lysed 16 hours after stimulation (Promega, Leiden, The Netherlands). Thereafter, an equal amount of BrightGlow was added and luciferase was measured immediately with a microplate reader (BMG, Isogen life science, De Meern, The Netherlands). Constitutively GFP expression by the CAGA-luc reporter was used to determine transduction efficiency and to control/compensate for TGFβ-unrelated biological and technological variations.

### Osteogenic and chondrogenic differentiation

Human subcutaneous adipose tissue from healthy male donors, with an age range of 33–47 years, was obtained from the Department of Plastic Surgery (Radboudumc) after ethical approval (Commissie Mensgebonden Onderzoek; dossier number #3252) and informed consent from 2017 to 2021. The tissue used for this study was collected in November 2017 and excised into pieces and then minced in a 0.1% collagenase type II (Mannheim, Germany) solution with 1% bovine serum albumin (BSA, Sigma, St. Louis, USA) for 1 h at 37°C under shaking conditions, as described previously [24]. Initially, cells were treated with FBS or four different EVdFBS (5%) media for 96 hours. After that, cells were washed carefully with PBS, and osteogenic differentiation was induced using with regular osteogenic induction media (FBS media supplemented with 20mM β-glycerophosphate, 100nM dexamethasone, and 50µM ascorbic acid phosphate) for 21 days. Cells were washed with PBS twice and fixed with 70% ethanol for 10 min, and stained with 0.5% alizarin red S (pH 4.1) for 20 min and then washed three times with deionized water.

As for chondrogenic differentiation, a micromass culture system was employed [25]. Cells were seeded at a concentration of $10^7$ cells/ml in the 24-well plate. The first 2 hours, the plates were incubated at 37°C with 5% $CO_2$ without extra culture media. Then cells were washed with PBS and treated with FBS or four different EVdFBS (5%) media. After 96 hours, chondrogenic induction medium (CIM, high glucose media supplemented with 10 ng/ml transforming growth factor-1, 1µM dexamethasone, 0.2 mM ascorbate-2-phosphate, 1mM sodium pyruvate and 1:100 diluted ITS + Premix) were replaced every other day to induce chondrogenic differentiation. After 14 days of induction, cells were lysed for total RNA extraction.

### EV treatment

The human knee synovial fibroblasts were isolated from the synovial biopsies taken from patients during joint replacement surgery at the orthopedics department of the Sint Maartenskliniek, Nijmegen, The Netherlands. This anonymous material was considered surgery surplus material and therefore, its use did not require ethical approval [26]. The synovial samples were digested using 50 µg/ml Liberase TM (Roche, Basel, Switzerland) for 1 h at 37°C in Roswell Park Memorial Institute (RPMI) culture medium without supplementations. The digestion was stopped by adding 10% fetal calf serum (FCS). Subsequently, the synovial cells were passed through a 70-µm cell strainer (Corning, NY, USA) and centrifuged. All cell centrifugations were performed for 5 min at 1500 rpm/423 g in a Heraeus Megafuge 16R (Thermo Fisher Scientific). Red blood cells were lysed for 2 min at RT using 4 ml RBC lysis buffer (155 nM NH4Cl, 12 mM KHCO3, 0.1 mM EDTA, pH 7.3). The lysis reaction was quenched by adding 6 ml RPMI culture medium, supplemented with 10% FCS and 1 mM pyruvate and 1% P/S.

To investigate the role of ADSC-EVs on collagen metabolism, the fibroblasts were treated with $2*10^8$ particles/ml EVs enriched from ADSCs FBS media, ADSCs-PEGEVdFBS media, ADSCs-serum free media or FBS-EVs. The cells were seeded at $10^5$ cells in 24-wells plates, and incubated for 12h in serum-free RPMI medium with EVs enriched from hADSCs-PEG EVdFBS media. The EVs enriched from hADSCs-FBS growth media and hADSCs-serum free media as well as vehicle (PBS) were deemed as the references. The cells were collected for subsequent total RNA extraction.

### RNA extraction and real-time PCR

RNA extraction from cells was performed using TRI reagent according to the manufacturer's procedure. The cells were mixed with 0.5 mL TRI reagent and 100μl chloroform, vortexed for 15 seconds, then incubated at room temperature for 2 minutes. The supernatant was transferred to a new tube and 250μl isopropanol was added after centrifugation at 12,000g for 15 minutes at 4˚C. The mixture was centrifuged at 12,000g for 30 minutes at 4˚C to remove the supernatant after an overnight incubation, and the RNA pellet was washed twice with 75% ethanol. After centrifugation at 12,000g for 5 minutes at 4˚C, the ethanol was aspirated and the RNA pellet was air dried for 10 minutes, and resolved in 8 μl of RNase-free water.

Synthesis of cDNA was accomplished by reverse transcription PCR an oligo(dT) primer and Moloney murine leukemia virus Reverse Transcriptase. Quantitative real-time PCR was performed using SYBR Green real-time PCR master mix on a Step-One according to the manufacturer's instructions. Primer sets for individual genes were used (S1 Table).

### Statistical analysis

The graphs (Figs 1B–1D, 2A–2C, 2E, 2F, 3B–3D, 3F–3H and 4) show the biological and/or technical replicates (n = 3 or 4). Statistical analyses were performed using GraphPad Prism 9 (GraphPad Software Inc., CA) statistical software. For XTT assay, CAGA12-Luciferase reporter assay, Brown-Forsythe and Welch one way ANOVA with Dunnett T3 multiple comparisons tests was used for statistical analysis, and $p<0.05$ or less was deemed significant.

## Results

### Comparison among EV depletion methods

Diluting FBS prior to performing UC is required to improve EV-depletion efficiency. The EV protein content (fg/particle), a purity index of the isolated FBS EV samples [27], was the lowest in UC isolated FBS EVs, isolated from 25% diluted FBS. This coincided with a marked reduction in particle numbers of UC EVdFBS batch. Moreover, comparable proliferations of hADSC were observed in FBS and UC EVdFBS, obtained through performing UC of 25% diluted FBS.

By performing UF of FBS, though more particles were removed from FBS compared to that by UC (Fig 1C), the purity of UF isolated FBS EVs was much worse as higher protein content was detected in UF isolated FBS EVs (Fig 1B). Meanwhile, the proliferation of hADSC was significantly decreased when using UF EVdFBS as culture media supplement.

Using different final concentration of PEG 6K for EVs isolation with a desirable effect on EVs precipitation, has been reported recently [28, 29]. To choose the optimal one for high selectivity of FBS-EVs depletion, a final incubation concentration in the range of 16%-10% PEG in FBS was investigated. We found an inverse relationship between the concentration of PEG and the particle number of batches and purity of isolated EVs. More importantly, though the amount of particle depletion was almost comparable between PEG precipitation and UF (Fig 1C), (concentration 10%) PEG EVdFBS could support hADSC growth (Fig 1D).

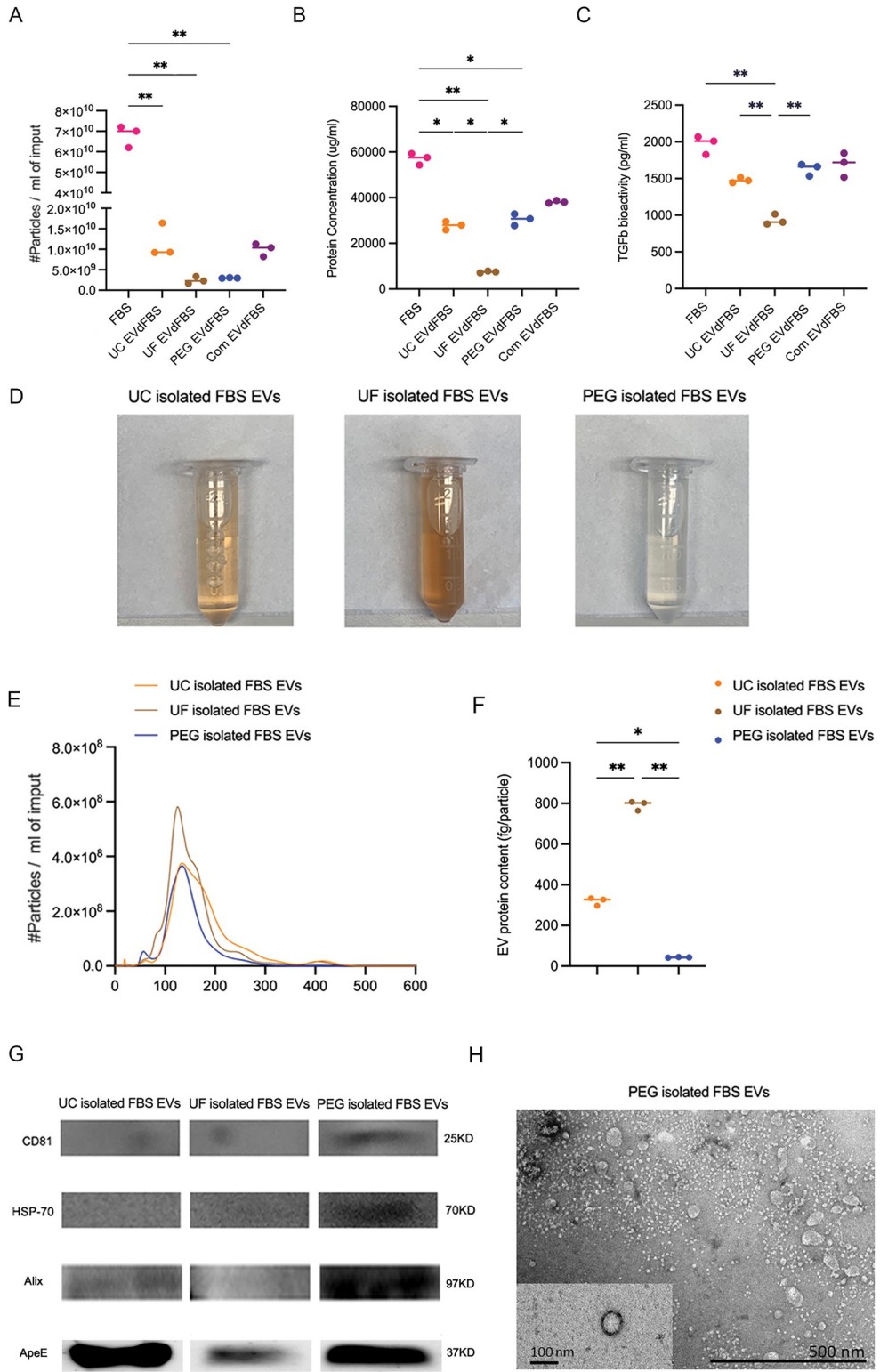

**Fig 2. Comparison among five FBS batches and three isolated FBS EVs samples.** (A) Particle concentration was determined by the NTA (in triplo). (B) Protein concentration was measured using micro-BCA kit (in triplo). (C) TGF-β activity was determined using CAGA-LUC reporter assay (in triplo). (D) Colors of isolated FBS EVs by UC, UF, and PEG (n = 3, in independent experiments). (E) Particle size distribution of UC, UF, and PEG isolated FBS EVs was determined using NTA (in triplo). (F) EV protein content was based on the ratio of the particle concentration over

protein content, determined respectively by NTA and microBCA. (G) Detection of EV-markers Alix, CD81 and HSP70 as well as the lipoprotein maker ApoE by western blotting in UC, UF, or PEG isolated FBS EVs. Statistical differences were determined by one way ANOVA tests. Values are mean, *$p<0.05$, **$p<0.005$.

Based on the results above, UC method using 25% diluted FBS and 10% PEG addition to FBS were selected for further analysis.

## PEG had the highest selectivity of FBS-EVs depletion

Compared to FBS, the amount of particle depletion was almost comparable between PEG (95.6%) and UF (96.6%) but less by UC (82%) (**Fig 2A**), but FBS protein loss was markedly different among PEG (47%), UF (87%), and UC (51%) (**Fig 2B**), therefore the ratio of EV depletion over protein loss was PEG (2.03), UC (1.61), and UF (1.11). A thorough EV depletion was observed by UF and PEG precipitation, and much protein aggregate in the FBS [30] was co-isolated with FBS-EVs by UF, leading the lowest total protein concentration in UF EVdFBS. Typically, compared by FBS group, a significantly decreased TGFβ/Smad signalling induced by UF EVdFBS was determined using CAGA12- luciferase reporter construct (**Fig 2C**). Quantitative values of the above results were shown in **Table 1**. We speculated that compared by UC, more particles were depleted but slightly less protein loss by PEG precipitation, implying the not only EVs but rich non-particle proteins including protein aggregates as well as lipoproteins exist in the FBS. Of note, the commercial EV depleted FBS (Com EVdFBS) exhibited comparable amount of particle concentration, protein content and TGFb bioactivity with UC EVdFBS and PEG EVdFBS.

In addition, we characterized the isolated FBS EVs in more detail to determine the best depletion method. PEG isolated FBS EVs showed less serum colour, compared with other two (**Fig 2D**). A less size variation of nanoparticles (30-300nm) was seen in PEG isolated FBS EVs (**Fig 2E**), with lower EV protein content (**Fig 2F**). Quantitative values of the above results were shown in **Table 2**. Moreover, only in the PEG isolated FBS EVs, Alix, CD81 and HSP-70 were clearly detectable. UC isolated FBS EVs had higher expression on ApoE, one marker of high-density lipoproteins (1.063–1.21 g/ml) that overlap in density with EVs (1.10–1.19 g/ml), compared with PEG did (**Fig 2G**), showing it is inevitable that lipid loss happened during UC and PEG precipitation. Notably, though the much protein aggregation loss happened by performing UF due to its low selectivity, the faint band of ApoE was observed in UF isolated FBS EVs, meaning some high-density lipoproteins (molecule weight below 100 kDa) could be retained in UF EVdFBS. Above all, the results implied PEG precipitation had the highest selectivity of FBS-EVs depletion among all techniques.

## PEG EVdFBS media supported hADSC growth, differentiation and enrichment of purer hADSC-EVs

The microscopic morphology of hADSC cultured in FBS and the four EVdFBS (5%) media was observed after 72h incubation. The low confluency of cells was observed in UF EVdFBS media (**Fig 3A**). Consistently, the colorimetric absorbance for proliferation as well as the mRNA level CCND1 and SIRT1 that genes important for cell growth were decreased UF EVdFBS media (**Fig 3B–3D**). To evaluate whether the ADSCs retained their differentiation capacity, the alizarin Red S staining assay was done and result exhibited the comparable calcium deposit in all cell layers after 96 hours expose in FBS and four EVdFBS media and subsequent regular osteogenic differentiation lasting for 21 days (**Fig 3E**). Lightly lower mRNA expression levels of Col2a were observed in all EVdFBS media during chondrogenic

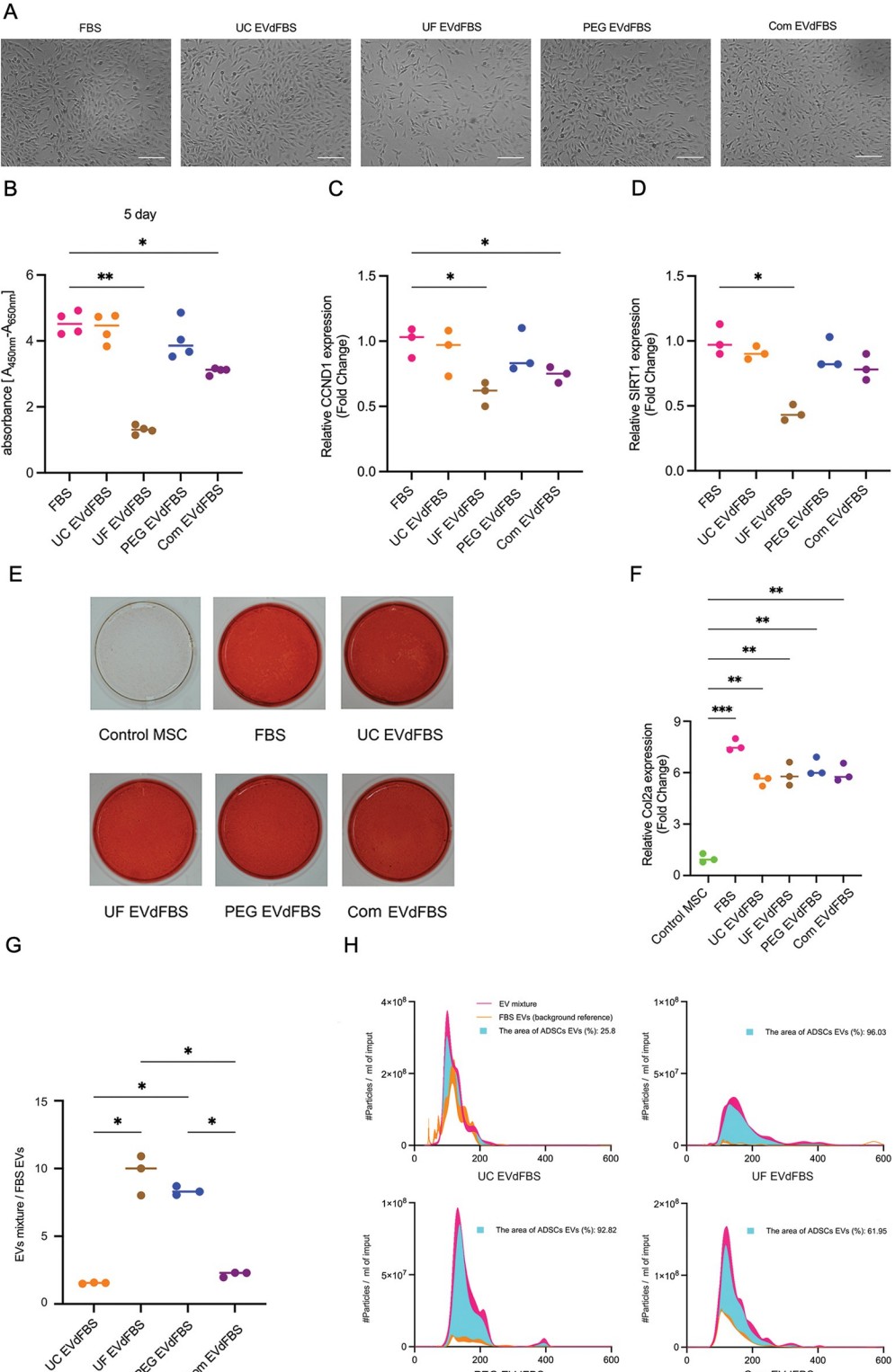

**Fig 3. Growth, differentiation and EV-release of human ADSCs cultured in five different FBS batches.** (A) Microscopic morphology of human ADSCs cultured in 5% FBS or four different EVdFBS batches. (B) Proliferation of ADSCs cultured in 5% FBS and four EVdFBS batches for 5 days determined using XTT assay (in quadruple). (C-D) CCND1, SIRT1 mRNA level of human ADSCs exposed to 5% FBS or four different EVdFBS batches were determined (fold change) using RT-qPCR (in triplo). (E) Osteogenic differentiation observed by alizarin Red S staining after

ADSCs cultured in 5% UT FBS or four different EVdFBS batches for 96 hours (in triplo). (F) Determined chondrogenic differentiation marker (Col2a) in MSC cultured in different FBS batches by RT-qPCR (in triplo). (G) The ratio of particle concentration of ADSC EVs to FBS-EVs (in triplo). (H) Particle size distribution was determined using NTA (in triplo). For panel B-D and F-G, statistical differences were determined by one way ANOVA tests. Values are mean, *p<0.05, **p<0.005, ***p<0.001.

differentiation (**Fig 3F**). As for enrichment of EVs, purer ADSC EVs could be obtained from hADSC-UF EVdFBS and hADSC-PEG EVdFBS media according to the higher ratio of EVs present in hADSC-EVdFBS media (EVs mixture) over the corresponding cell-free EVdFBS media (FBS-EVs) (**Fig 3G**). Furthermore, the area of size distribution exhibited more ADSC EVs were in the PEG EVdFBS media, compared to those in hADSC-UF EVdFBS media (**Fig 3H**).

## EVs enriched from ADSCs-PEG EVdFBS media positively regulate collagen metabolism

To determine whether EVs enriched from ADSC-PEG EVdFBS media retained their tissue regenerative potential, 12-hour serum-free expose with EVs stimulation was conducted. Quantitative real-time RT-PCR showed that the mRNA levels of Col1a1 and TIMP1 of human knee synovial fibroblasts were increased significantly with EVs stimulation, compared to those with vehicles (**Fig 4**).

## Discussion

Recent publications have investigated some FBS-EV depletion methods regarding cellular physiology state and function [10, 22]. Nevertheless, barely any specific attention has been paid on depletion efficiency of these methods through evaluation of isolated FBS-EVs sample

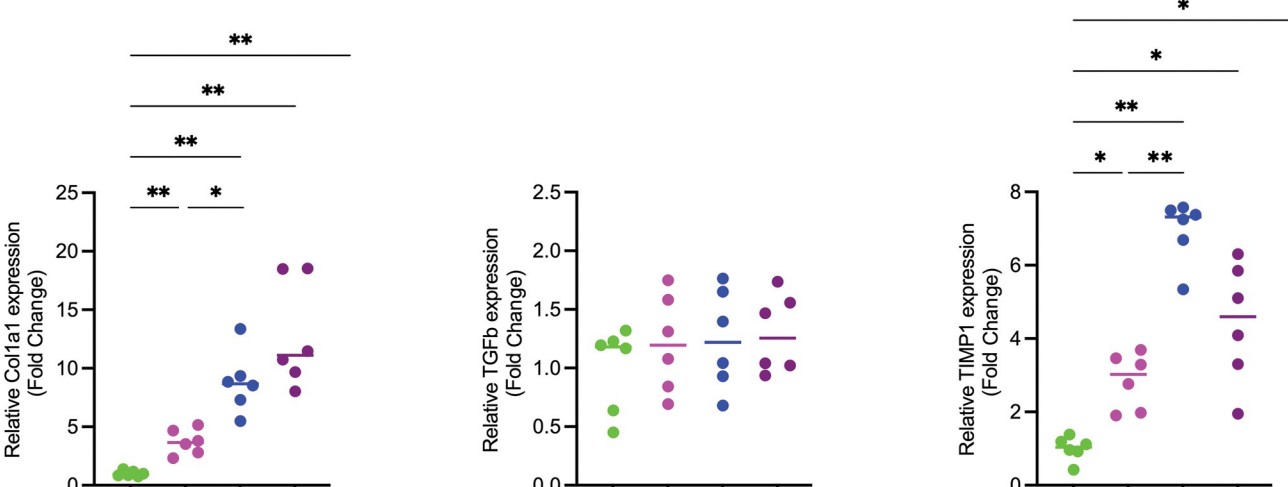

**Fig 4. Gene expression associated with collagen metabolism.** Col1a1, TGFβ, TIMP1 mRNA levels of human knee synovial fibroblast exposed to different sources of EVs were determined (fold change) using RT-qPCR (n = 6). Statistical differences were determined by one way ANOVA tests. Values are mean, *p<0.05, **p<0.005.

**Table 1. EVdFBS characterization.**

| Sample | Particles/ml of imput | Protein Concentration (ug/ml) | TGFb activity (pg/ml) |
|---|---|---|---|
| FBS | $6.80 \times 10^{10}$ | 57063±2562 | 1968±126 |
| UC EVdFBS | $1,16 \times 10^{10}$ | 27792±1821 | 1478±36 |
| UF EVdFBS | $2.43 \times 10^{9}$ | 7435±389 | 934±72 |
| PEG EVdFBS | $2.99 \times 10^{9}$ | 30472±2548 | 1629±83 |
| Com EVdFBS | $1.11 \times 10^{10}$ | 38140±580 | 1694±165 |

and EVs enriched from cell EVdFBS media. In this study, we demonstrated PEG precipitation is the optimal method to deplete EVs in FBS. Due to its high selectivity of EVs depletion, PEG EVdFBS could be used as culture media supplement for MSC culture to obtain purer MSC-EVs.

FBS-EVs have been continuously reported to be bioactive as they can be internalized by the recipient cells and influence cellular physiology, such as cell migration [18, 31], inevitably disturbing the analysis of cargo content and the biological properties of EVs released by the cultured cells. Of note, co-isolated non-EV particles could be removed from EVs mixture through further wash by sucrose gradient ultracentrifugation and SEC isolation [32], but separating two kinds of EVs (FBS-EVs and cell-EVs) directly is impossible with the available isolation techniques [30]. Therefore, making FBS devoid of bovine EVs with minimal loss of serum protein is essential when using it as a culture medium supplement to maintain cell growth and performance and enrich pure cell-EVs.

The most commonly used method to obtain EV dFBS is UC, but our study showed that this is not only a time-consuming method but also insufficient to remove FBS-EV. By UF and PEG, comparable amount of particle depletion was achieved, but the higher EV protein content in UF isolated FBS-EVs sample and reduced ADSCs proliferation in UF EVdFBS media implied UF had low selectivity of FBS-EV depletion. Considering FBS contains proteins, growth factors, as well as lipids, vitamins, carbohydrates, all of which are essential for cell attachment and maintenance, we speculated much protein loss in UF EVdFBS was one of the potential cause UF EVdFBS influenced morphology and barely support growth of hADSC [33]. Moreover, it is essential and desirable to make an assessment on the isolated FBS-EVs sample through a series of EV characterization parameters [34]. The serum color caused by bilirubin, which is transported as an albumin complex in blood under normal condition [35], were deeper in the isolated FBS EVs obtained by UC and UF implied the presence of more non-EV particles like albumin aggregates. The EV specific markers Alix, CD81, HSP-70 were only clearly detectable in PEG isolated FBS-EVs, which meant purer FBS-EVs were isolated by PEG precipitation. Interestingly, though the much essence loss happened by UF due to its low selectivity, the faint band of ApoE was observed in UF isolated FBS EVs, meaning some high-density lipoproteins (molecule weight below 100 kDa) without aggregation could be retained in UF EVdFBS. Of note, it was difficult to explain the difference with the positive results observed by Kornilov and colleagues. The UF protocol might not be a suitable method on FBS

**Table 2. Isolated FBS EVs characterization.**

| Sample | Particles/ml of imput | NTA size (mode) | EV protein content (fg/particle) |
|---|---|---|---|
| UC | $1.24 \times 10^{11}$ | 136.9±10 | 319±20 |
| UF | $1.79 \times 10^{11}$ | 125.1±3 | 791±24 |
| PEG | $1.08 \times 10^{11}$ | 129.5±9 | 43±1.5 |

EV depletion in our study. we meticulously replicated the UF procedure outlined in Kornilov and colleagues' work, employing the identical UF product (UFC910024, 100K Merk Millipore Ltd). As anticipated, our findings aligned with their study, demonstrating a similar degree of EV depletion (approximately 96% in Kornilov's findings). Nevertheless, in their study despite observing diminished cellular dimensions of human adipose stem cells cultivated in UF EV depleted FBS media group over 96 hours, as evidenced by microscopic analysis, the metabolic activity per cell exhibited no statistically significant divergence from the FBS control group in their study. In our view, the divergent experimental outcomes were attributed to the variation in FBS concentration (5% versus 10%), and by using 10% they might have compensated for the loss of growth factors using the UF method.

Although hADSC still possessed differentiation ability after 96 hours expose in four EVdFBS media, a slightly lower expression of Col2a was detected in four EVdFBS media compared to that in FBS media, which implied equal differentiation condition for 14 days could not completely diminish the difference of chondrogenic differentiation capacity between hADSC in FBS media and in four EVdFBS media. It seemed the initial 96-hour EVdFBS exposure affected chondrogenic differentiation ability of hADSCs. Indeed, it was reported that serum-supplementation could increase the frequency of chondrogenic cells and ECM accumulation to enhance chondrogenesis of aging MSCs [36]. Therefore, using serum-free culture media for MSCs culture and EVs enrichment might cause cells to go into adaptation and survival mode, and change the cargo of their EVs and therapeutic property, making further research inaccurate [37, 38]. In addition, a large scale of MSC-EVs production for clinical application requires sustainable and normal MSC culture condition [39]. Using EV-depleted culture media is a standard and specific option, which has no impact on persistent use of MSC.

The considerable proportion that FBS-EVs were in EVs mixture implied the highly impure ADSC-EVs were enriched from ADSC-UC EVdFBS and ADSC-Com EVdFBS media. Furthermore, EVs mixture in ADSC-UF EVdFBS media had broader particle size range, which might influence property of EVs and analysis at a functional level of EVs-mediated intercellular signaling [40, 41]. Moreover, we speculated that the genes associated with EV production and trafficking (e.g.VPS4A) [42] were affected when hADSCs were cultured in TGFβ deficient condition [43], causing less distinct peak of EVs mixture.

It was reported that EVs derived from ADSCs potentiating tissue repair [9, 44, 45]. Of note, the UC EVdFBS media and serum-free media were used for supporting ADSCs growth and EVs enrichment in these publications. Therefore, we treated knee synovial fibroblast with EVs derived from ADSC-FBS media, ADSC-PEG EVdFBS media, ADSC-serum free media. FBS-EVs were used as a reference. Based on the published study [46, 47], the mRNA levels of Col1a1, TIMP1 in vaginal fibroblasts/ scleral fibroblasts were determined to evaluate the collagen metabolism. In our study, the results showed EVs enriched from ADSC-PEG EVdFBS media participate in the processes of extracellular matrix remodeling, mainly collagen metabolism. Interestingly, higher mRNA levels of Col1a1 and TIMP1 were expressed in EVs enriched from ADSC-PEG EVdFBS media and ADSC serum free media groups, compared to that from ADSC-FBS media group. It seemed a competitive relationship exists between FBS-EVs and cell-EVs, influencing the therapeutic use of cell-EVs. Of note, Azadeh *et al* demonstrated the effective particles could be removed from FBS through PEG 4K precipitation, with 3.2% PEG in final solution [48]. However, we selected the common concentration range of PEG employed in previous literature, and in our study the particles in PEG EVdFBS have increased following the decrease of final solution, which is likely to be ascribed to different commercial FBS. In terms of the status of cells cultured in PEG EVdFBS, the authors observed that the primary adipose tissue derived cells cultured in 10% PEG 4K mediated EVdFBS expressed CD44 (99.7%), CD105 (99.8%) and CD29 (97.2%), as markers for stem cells. While we focused on

the differentiation capacity of cells, which was not affected. Moreover, in our study the cell-derived EVs obtained from PEG EVdFBS supernatant were obtained and still retained bioactive, meaning the PEG EVdFBS supports cell-EVs enrichment.

For clinical use, utilizing MSC EVs necessitates an expansion of EV production on a larger scale, adherence to rigorous good manufacturing practice (GMP) protocols [39], and other various regulatory requisites. In this context, our PEG precipitation depletion protocol emerges as a noteworthy advancement. This protocol enables the generation of FBS EV-free serum, effectively supporting both the production and inherent properties of cell derived EVs. More importantly, because of the FDA approved safety, PEG precipitation holds promising applicability to a wide range of serums, including human serum. As a result, it presents a compelling avenue for future GMP-compliant EV production tailored to clinical applications in the realms of regenerative medicine and therapy.

As one routine EV isolation method, emerging research about PEG precipitation has been conducted recently [28, 49]. In this study, we demonstrated the relatively high FBS-EV selectivity of PEG precipitation, while the underlying mechanism remains unclear. Therefore, it should pay more attention on study the effect of PEG on the sorting of FBS-EVs as well as EVs from other sources in the future. In addition, the results in our study showed that PEG had no influence on the functional effect of cell-derived EVs, while in terms of the physiology/pathology process that EVs participate in, more studies about the effect of PEG on EVs transmission and uptake by target cells deserve to be explored.

## Limitation

In our investigation, our primary focus centered on FBS-derived extracellular vesicles (EVs) and protein contents stemming from FBS contaminants. Considering that PEG is non-toxic, non-immunogenic, non-antigenic, highly soluble in water, and FDA approved, we don't expect the presence of PEG have damage effect on cells, but we have not checked that if potential PEG contamination has effects on the cell-derived EVs biogenesis, especially the protein content composition.

In addition, we are aware of that MSCs have been isolated from bone marrow or adipose tissue, as well as skin and foreskin [50], synovial fluid [51], Wharton's jelly [52] for specific tissue regeneration. The common cell surface markers of BMSCs or ADSCs or Wharton's jelly MSCs are CD73, CD90, CD105, while CD44 is present on MSCs from skin and synovial fluid. While regarding the similar therapeutic use and effects of MSC-EVs, which are capable of restoring an extensive range of damaged or diseased tissues and organs, we selected one typical MSCs as research object [53].

In conclusion, our research has successfully demonstrated the straightforward preparation of PEG EVdFBS within any conventional research laboratory setting. The PEG precipitation method we've developed offers a standardization process. Notably, it facilitates the sustained proliferation of MSCs for a minimum of 5 days. The integration of our PEG precipitation protocol within the broader MSC EV research community holds great potential. By adopting this method, researchers can acquire pure MSC EVs, enhancing the robustness and reliability of their investigations. In effect, the quality of their research endeavors stands to benefit significantly. In light of these advancements, PEG precipitation emerges as a better method than conventional UC as well as UF for the purpose of growing MSCs and obtain pure MSC-derived EVs not contaminated with FBS-EVS.

## Supporting information

**S1 Table. Sequence of primers for real-time PCR analysis.**
(DOCX)

**S1 File. Characterization for isolated FBS EVs.**
(ZIP)

## Acknowledgments

We thank Onno J. Arntz for his technical assistance, Prof. Jeroen for critical reading and English correction of the manuscript, and the Radboud Institute for Molecule Life Sciences, Radboud University Medical Center. Peng Wang is grateful for scholarship from the China Scholar Council, and thank Peter ten Dijke who kindly provided us with adenoviral CAGA12-luc construct.

## Author Contributions

**Conceptualization:** Peng Wang, Van der Kraan P. M., Jeroen J. J. P. van den Beucken, Fons A. J. van de Loo.

**Investigation:** Peng Wang, Onno J. Arntz.

**Methodology:** Jeroen J. J. P. van den Beucken, Fons A. J. van de Loo.

**Project administration:** Peng Wang, Johanna F. A. Husch.

**Supervision:** Fons A. J. van de Loo.

**Writing – original draft:** Peng Wang.

**Writing – review & editing:** Van der Kraan P. M., Jeroen J. J. P. van den Beucken, Fons A. J. van de Loo.

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
