## [Decision Letter · Decision Letter 0]

18 Jul 2023

PONE-D-23-13193Polyethylene Glycol Precipitation is an Efficient Method to Obtain Extracellular Vesicle-Depleted Fetal Bovine SerumPLOS ONE

Dear Dr. Wang,

Thank you for submitting your manuscript to PLOS ONE. After careful consideration, we feel that it has merit but does not fully meet PLOS ONE’s publication criteria as it currently stands. Therefore, we invite you to submit a revised version of the manuscript that addresses the points raised during the review process.

We look forward to receiving your revised manuscript.

Kind regards,

Wilfried A. Kues, Ph.D.

Academic Editor

PLOS ONE

Journal Requirements:

3. In the methodology you have mentioned that samples were collected in 2017. Please specify the date range when sample collection was done.

"NO"

"NO"

7. PLOS requires an ORCID iD for the corresponding author in Editorial Manager on papers submitted after December 6th, 2016. Please ensure that you have an ORCID iD and that it is validated in Editorial Manager. To do this, go to ‘Update my Information’ (in the upper left-hand corner of the main menu), and click on the Fetch/Validate link next to the ORCID field. This will take you to the ORCID site and allow you to create a new iD or authenticate a pre-existing iD in Editorial Manager. Please see the following video for instructions on linking an ORCID iD to your Editorial Manager account: https://www.youtube.com/watch?v=_xcclfuvtxQ

Reviewers' comments:

Reviewer's Responses to Questions

**Comments to the Author**

1. Is the manuscript technically sound, and do the data support the conclusions?

Reviewer #1: Partly

2. Has the statistical analysis been performed appropriately and rigorously? 

Reviewer #1: No

3. Have the authors made all data underlying the findings in their manuscript fully available?

Reviewer #1: No

4. Is the manuscript presented in an intelligible fashion and written in standard English?

Reviewer #1: No

5. Review Comments to the Author

Reviewer #1: Major Comments:

Abstract:

1. The abstract's clarity can be enhanced by providing a rationale or explanation for the necessity of EV-free FBS in MSC culture.

2. The abstract compares three methods (UC, UF, and PEG) of obtaining EV-depleted FBS but does not explicitly state the preferred method in current MSC research. It would be beneficial to clarify this.

3. The abstract needs a brief explanation of the importance of TGFβ/Smad signaling in MSC biology to highlight the significance of the observed decrease in TGFβ/Smad signaling via the UF method.

4. The abstract should include more detail about the different EVdFBS (5%) media used and how they influenced the differentiation capacity of hADSCs.

5. The nature of the significant effect on cell proliferation in UF EVdFBS media should be clarified.

Introduction:

1. The introduction should mention other commonly used techniques for EV depletion to provide comprehensive context.

2. The introduction could benefit from further elaboration on the specific limitations of the tested EV-depletion methods and how these impact the quality of the EV-depleted FBS.

Materials and Methods:

1. The section should include more specific details on ethical approval and informed consent.

2. The number of replicates and independent experiments for each assay should be specified to understand the statistical analysis better.

3. More explicit mention of positive and negative controls and appropriate standards is necessary to understand the experimental setup better.

4. More specific information about the statistical tests used for specific analyses or comparisons is needed.

Results:

1. The Results section should include statistical tests and report p-values for the observed differences between experimental groups.

2. Replication information should be provided, like the number of replicates or independent experiments conducted.

3. The Results section should include more thorough comparisons and discussions of the results.

Discussion:

1. The discussion should address the limitations or potential shortcomings of the study.

2. A clearer comparison or discussion of how the current findings align with or differ from previous literature on FBS-EV depletion methods is needed.

3. Some speculative statements in the discussion lack supporting evidence or references and should be backed up with appropriate data or citations.

4. The discussion should more thoroughly interpret the functional implications of the findings.

5. A more detailed discussion of the clinical implications of the study's findings is needed.

6. The discussion should include suggestions for future research directions or practical applications of the results.

Minor Comments:

Abstract:

1. The abbreviations used in the abstract should be defined upon their first mention or provided in a list for reference.

Materials and Methods:

1. Details on how the data were presented and represented would enhance the section.

2. Information on the availability of specific reagents, kits, or equipment used in the study would aid in the study's reproducibility.

Results:

1. The Results section should report actual values, means, standard deviations, or other relevant statistical measures to enhance understanding of the effects observed.

Discussion:

1. The discussion could be more complete by offering a more comprehensive and insightful interpretation of the study's results, considering limitations, and providing a broader discussion of implications.

6. PLOS authors have the option to publish the peer review history of their article (what does this mean?). If published, this will include your full peer review and any attached files.

Reviewer #1: **Yes: **M V Sasidhar

---

## [Author Response · Author response to Decision Letter 0]

23 Sep 2023

1. The abstract's clarity can be enhanced by providing a rationale or explanation for the necessity of EV-free FBS in MSC culture.

Response: To maintain growth of MSCs and obtain pure MSC-derived EVs, the culture media should contain fetal bovine serum (FBS) devoid of EVs, as the presence of FBS-EVs confounds the properties of MSC-EVs.

2. The abstract compares three methods (UC, UF, and PEG) of obtaining EV-depleted FBS but does not explicitly state the preferred method in current MSC research. It would be beneficial to clarify this.

Response: 18h ultracentrifugation (UC) and ultrafiltration (UF), which are common FBS-EV depletion methods in current MSC-EV research.

3. The abstract needs a brief explanation of the importance of TGFβ/Smad signaling in MSC biology to highlight the significance of the observed decrease in TGFβ/Smad signaling via the UF method.

Response: A significant decrease of TGFβ/Smad signaling, involving in MSC growth and physiology, was observed by UF.

4. The abstract should include more detail about the different EVdFBS (5%) media used and how they influenced the differentiation capacity of hADSCs.

Response: After 96 hours of exposure to 5% FBS or 5% four different EVdFBS cell growth media, the osteogenesis ability of hADSCs was not impaired but slightly lower mRNA expression level of Col2a observed in EVdFBS media during chondrogenesis. 

5. The nature of the significant effect on cell proliferation in UF EVdFBS media should be clarified.

Response: In consistent with low confluency of hADSCs observed by optical microscope, cell proliferation in response to 5% UF EVdFBS media was inhibited significantly. 

6. The introduction should mention other commonly used techniques for EV depletion to provide comprehensive context.

Response: Yes, thank you for your suggestion. And now we changed it accordingly. 

Of note, considering retrieve of a large scale of FBS volumes, methods used to deplete the EVs form FBS should be a single step procedure that can be used to process large volumes, and methods such as size exclusion chromatography (SEC isolation), or affinity-based isolation methods are either costly or not applicable. A drawback of using commercially available EV-depleted FBS is that the method is not made public. Given that certain studies have revealed contaminants in commercial EV-free FBS, caution should be exercised when using such material for in vitro EV studies

7. The introduction could benefit from further elaboration on the specific limitations of the tested EV-depletion methods and how these impact the quality of the EV-depleted FBS.

Response: Shelke and colleagues conducted a study on EV depletion protocol using different centrifugation times for FBS and observed that 18 h of UC was the optimal time, even though it did not completely eliminate EV contaminants from FBS. Pham et al later challenged the UC protocol and investigated the particles remaining in FBS after EV depletion through UC, showing that certain nanoparticles were not depleted after 18 h of UC. For UF FBS EV depletion, Kornilov et al filtered FBS through ultra-15 centrifugal filters (55 minutes at 3000 × g) to obtain UF EV depleted FBS. Transmission electron microscopy revealed UF EV depleted FBS had no detectable EVs, distinguishing it from commercially EV-depleted FBS and UC EV depleted FBS. In addition, they observed UF EVdFBS contained no any protein, but a distinct protein pattern was detected in UF isolated FBS EVs in the study. Therefore, current techniques not only incompletely deplete EVs but also may remove essential FBS proteins. During the FBS-EVs depletion process, the serum factors (proteins, lipids, and lipoproteins) tend to become aggregated and removed from FBS, leaving a less potent and incomplete cell culture media supplement.

8. The section should include more specific details on ethical approval and informed consent.

Response: Human subcutaneous adipose tissue from two healthy male donors, with an age range of 33–47 years, was obtained from the Department of Plastic Surgery (Radboudumc) after ethical approval (Commissie Mensgebonden Onderzoek; dossier number #3252) and informed consent. As for the human knee synovial fibroblast, the human knee synovial fibroblasts were isolated from the synovial biopsies taken from patients during joint replacement surgery at the orthopedics department of the Sint Maartenskliniek, Nijmegen, The Netherlands. This anonymous material was considered surgery surplus material and therefore, its use did not require ethical approval (A three-dimensional model to study human synovial pathology, ALTEX.2019;36(1):18-28. doi: 10.14573/altex.1804161. Epub 2018 Oct 9.)

9. The number of replicates and independent experiments for each assay should be specified to understand the statistical analysis better.

Response: Thank you for your attentive observation and they have been added into Figure Legend part. 

10. More explicit mention of positive and negative controls and appropriate standards is necessary to understand the experimental setup better.

Response: You are correct and more details have been added into Methods and Materials part. 

NTA assay: Particles were diluted in PBS, till a suitable concentration for analysis was reached, and the concentration value between 20 to 80 particles/frame was deemed as effective measurement.

Protein content: the 2mg/ml BSA was the standard protein sample.

XTT assay: The assay was done according to the manufacturer's instructions. After culturing hADSCs for 48 hours or 5 days, cellular metabolic activity was determined. By adding XTT label reagent, absorbance was measured after incubation for 4hr at 37*C at wavelength 450 nm using a microplate reader. The blank well on the same row and the wavelength >650 nm were taken as the references.

Western blot: equal amounts of protein were loaded onto 12% SDS-PAGE, together with protein ladder.

CAGA12-Luciferase reporter assay: Constitutively GFP expression by the CAGA-luc reporter was used to determine transduction efficiency and to control/compensate for TGFβ-unrelated biological and technological variations.

EV treatment: The EVs enriched from hADSCs-FBS growth media and hADSCs-serum free media as well as vehicle (PBS) were deemed as the references.

11. More specific information about the statistical tests used for specific analyses or comparisons is needed.

Response: Thank you for pointing us at this shortcoming. The graphs (Figures 1B-1D, 2A-C, 2E-2F, 3B-D, 3F-H, and Figure 4) show the biological and/or technical replicates (n=3 or 4). Statistical analyses were performed using GraphPad Prism 9 (GraphPad Software Inc., CA) statistical software. For XTT assay, CAGA12-Luciferase reporter assay, Brown-Forsythe and Welch one way ANOVA with Dunnett T3 multiple comparisons tests was used for statistical analysis, and p＜0.05 or less was deemed significant. This information has been added to the manuscript.

12. The Results section should include statistical tests and report p-values for the observed differences between experimental groups.

Response: In fig 1, For panel C, the p was determined by student two tailed t test, values are mean. And for panels B and D were calculated by one way ANOVA tests. ** p<0.01. In fig 3, For panel B-D and F-G, statistical differences were determined by one way ANOVA tests. Values are mean, *p<0.05, **p<0.005, ***p<0.001. In fig 2 and fig 4, Statistical differences were determined by one way ANOVA tests. Values are mean, *p<0.05, **p<0.005. This information has been added to the figure legend part.

13. Replication information should be provided, like the number of replicates or independent experiments conducted.

Response: You are correct and they have been added into Figure Legend part. 

14. The Results section should include more thorough comparisons and discussions of the results.

Response: Yes, we agree, they have been included. For example, to explain result in fig 2B, “FBS protein loss was markedly different among PEG (47%), UF (87%), and UC (51%) (Fig 2B), therefore the ratio of EV depletion over protein loss was PEG (2.03), UC (1.61), and UF (1.11). A thorough EV depletion was observed by UF and PEG precipitation, and much more protein aggregates were co-isolated with FBS-EVs by UF, leading the lowest total protein concentration in UF EVdFBS”. To explain the results in fig 2, we speculated that compared by UC, more particles were depleted but slightly less protein loss by PEG precipitation, implying the not only EVs but rich non-particle proteins including protein aggregates as well as lipoproteins exist in the FBS.

15. The discussion should address the limitations or potential shortcomings of the study.

Response: We have rewritten the discussion accordingly. For the limitations: In our investigation, our primary focus centered on FBS-derived extracellular vesicles (EVs) and protein contents stemming from FBS contaminants. Considering that PEG is non-toxic, non-immunogenic, non-antigenic, highly soluble in water, and FDA approved, we don’t expect the presence of PEG have damage effect on cells, but we have not checked that if potential PEG contamination has effects on the cell-derived EVs biogenesis, especially the protein content composition. In addition, we are aware of that MSCs have been isolated from bone marrow or adipose tissue, as well as skin and foreskin, synovial fluid, Wharton's jelly for specific tissue regeneration. The common cell surface markers of BMSCs or ADSCs or Wharton's jelly MSCs are CD73, CD90, CD105, while CD44 is present on MSCs from skin and synovial fluid. While regarding the similar therapeutic use and effects of MSC-EVs, which are capable of restoring an extensive range of damaged or diseased tissues and organs, we selected one typical MSCs as research object.

16. A clearer comparison or discussion of how the current findings align with or differ from previous literature on FBS-EV depletion methods is needed.

Response: Yes, they have been added into the Discussion part. For example, “it was difficult to explain the difference with the positive results observed by Kornilov and colleagues. The UF protocol might not be a suitable method on FBS EV depletion in our study. we meticulously replicated the UF procedure outlined in Kornilov and colleagues' work, employing the identical UF product (UFC910024, 100K Merk Millipore Ltd). As anticipated, our findings aligned with their study, demonstrating a similar degree of EV depletion (approximately 96% in Kornilov's findings). Nevertheless, despite observing diminished cellular dimensions of human adipose stem cells cultivated in UF EV depleted FBS media group over 96 hours, as evidenced by microscopic analysis, the metabolic activity per cell exhibited no statistically significant divergence from the FBS control group in their study. In our view, the divergent experimental outcomes were attributed to the variation in FBS concentration (5% versus 10%), and by using 10% they might have compensated for the loss of growth factors using the UF method.” 

In addition, Azadeh et al demonstrated the effective particles could be removed from FBS through PEG 4K precipitation, with 3.2 % PEG in final solution. However, we selected the common concentration range of PEG employed in previous literature, and in our study the particles in PEG EVdFBS have increased following the decrease of final solution, which is likely to be ascribed to different commercial FBS. In terms of the status of cells cultured in PEG EVdFBS, the authors observed that the primary adipose tissue derived cells cultured in 10% PEG 4K mediated EVdFBS expressed CD44 (99.7%), CD105 (99.8%) and CD29 (97.2%), as markers for stem cells. While we focused on the differentiation capacity of cells, which was not affected. Moreover, in our study the cell-derived EVs obtained from PEG EVdFBS supernatant were obtained and still retained bioactive, meaning the PEG EVdFBS supports cell-EVs enrichment.

17. Some speculative statements in the discussion lack supporting evidence or references and should be backed up with appropriate data or citations.

Response: Yes, some citations have been added into the discussion part. For example, “Considering FBS contains proteins, growth factors, as well as lipids, vitamins, carbohydrates, all of which are essential for cell attachment and maintenance, we speculated much protein loss in UF EVdFBS was one of the potential causes UF EVdFBS influenced morphology and barely support growth of hADSC (Gstraunthaler G. Alternatives to the use of fetal bovine serum: serum-free cell culture. Altex. 2003;20(4):275-81. Epub 2003/12/13. PubMed PMID: 14671707).” 

“Moreover, we speculated that the genes associated with EV production and trafficking (e.g., VPS4A) (Aswad H, Jalabert A, Rome S. Depleting extracellular vesicles from fetal bovine serum alters proliferation and differentiation of skeletal muscle cells in vitro. BMC Biotechnol. 2016; 16:32. Epub 2016/04/04. doi: 10.1186/s12896-016-0262-0) were affected when hADSCs were cultured in TGFβ deficient condition (Lin H, Zhang R, Wu W, Lei L. miR-4454 Promotes Hepatic Carcinoma Progression by Targeting Vps4A and Rab27A. Oxidative medicine and cellular longevity. 2021; 2021:9230435. Epub 2021/11/16. doi: 10.1155/2021/9230435), causing less distinct peak of EVs mixture.”

18. The discussion should more thoroughly interpret the functional implications of the findings.

Response: Some discussion about functional implications have been added to the manuscript. “Our research has successfully demonstrated the straightforward preparation of PEG EVdFBS within any conventional research laboratory setting. The PEG precipitation method we've developed offers a standardization process. Notably, it facilitates the sustained proliferation of MSCs for a minimum of 5 days. The integration of our PEG precipitation protocol within the broader MSC EV research community holds great potential. By adopting this method, researchers can acquire pure MSC EVs, enhancing the robustness and reliability of their investigations. In effect, the quality of their research endeavors stands to benefit significantly. In light of these advancements, PEG precipitation emerges as a better method than conventional UC as well as UF for the purpose of growing MSCs and obtain pure MSC-derived EVs not contaminated with FBS-EVS.

19. A more detailed discussion of the clinical implications of the study's findings is needed.

Response: For clinical use, utilizing MSC EVs necessitates an expansion of EV production on a larger scale, adherence to rigorous good manufacturing practice (GMP) protocols, and other various regulatory requisites. In this context, our PEG precipitation depletion protocol emerges as a noteworthy advancement. This protocol enables the generation of FBS EV-free serum, effectively supporting both the production and inherent properties of cell derived EVs. More importantly, because of the FDA approved safety, PEG precipitation holds promising applicability to a wide range of serums, including human serum. As a result, it presents a compelling avenue for future GMP-compliant EV production tailored to clinical applications in the realms of regenerative medicine and therapy.

20. The discussion should include suggestions for future research directions or practical applications of the results.

Response: As one routine EV isolation method, emerging research about PEG precipitation has been conducted recently (Extracellular Vesicles of Mesenchymal Stem Cells Are More Effectively Accessed through Polyethylene Glycol-Based Precipitation than by Ultracentrifugation; Extracellular Vesicles of Mesenchymal Stem Cells Are More Effectively Accessed through Polyethylene Glycol-Based Precipitation than by Ultracentrifugation). In this study, we demonstrated the relatively high FBS-EV selectivity of PEG precipitation, while the underlying mechanism remains unclear. Therefore, it should pay more attention on study the effect of PEG on the sorting of FBS-EVs as well as EVs from other sources in the future. In addition, the results in our study showed that PEG had no influence on the functional effect of cell-derived EVs, while in terms of the physiology/pathology process that EVs participate in, more studies about the effect of PEG on EVs transmission and uptake by target cells deserve to be explored.

21. The abbreviations used in the abstract should be defined upon their first mention or provided in a list for reference.

Response: Yes, they have been defined in the abstract.

22. Details on how the data were presented and represented would enhance the section.

Response: Yes, they have been replenished.

23. Information on the availability of specific reagents, kits, or equipment used in the study would aid in the study's reproducibility.

Response: Yes, they have been replenished.

24. The Results section should report actual values, means, standard deviations, or other relevant statistical measures to enhance understanding of the effects observed.

Response: 

Table 1. EVdFBS characterization

Sample Particles/ml of imput Protein Concentration

(ug/ml) TGFb activity 

(pg/ml)

FBS 6.80×1010 57063±2562.0 1968±126.0

UC EVdFBS 1,16×1010 27792±1821.0 1478±36.4

UF EVdFBS 2.43×109 7435±389.2 934±72.1

PEG EVdFBS 2.99×109 30472±2548.0 1629±82.6

Com EVdFBS 1.11×1010 38140±580.3 1694±165.0

Table 2. isolated FBS EVs characterization

Sample Particles/ml of imput NTA size (mode) EV protein content 

(fg/particle)

UC 1.24×1011 136.9±10.0 319±19.8

UF 1.79×1011 125.1±2.6 791±24.0

PEG 1.08×1011 129.5±8.8 43±1.5 

25. The discussion could be more complete by offering a more comprehensive and insightful interpretation of the study's results, considering limitations, and providing a broader discussion of implications.

Response: It has been responded above.

---

## [Decision Letter · Decision Letter 1]

15 Nov 2023

Polyethylene Glycol Precipitation is an Efficient Method to Obtain Extracellular Vesicle-Depleted Fetal Bovine Serum

PONE-D-23-13193R1

Dear Dr. Wang,

We’re pleased to inform you that your manuscript has been judged scientifically suitable for publication and will be formally accepted for publication once it meets all outstanding technical requirements.

Kind regards,

Wilfried A. Kues, Ph.D.

Academic Editor

PLOS ONE

Additional Editor Comments (optional):

Reviewers' comments:

Reviewer's Responses to Questions

**Comments to the Author**

1. If the authors have adequately addressed your comments raised in a previous round of review and you feel that this manuscript is now acceptable for publication, you may indicate that here to bypass the “Comments to the Author” section, enter your conflict of interest statement in the “Confidential to Editor” section, and submit your "Accept" recommendation.

Reviewer #1: All comments have been addressed

2. Is the manuscript technically sound, and do the data support the conclusions?

Reviewer #1: Yes

3. Has the statistical analysis been performed appropriately and rigorously? 

Reviewer #1: Yes

4. Have the authors made all data underlying the findings in their manuscript fully available?

Reviewer #1: Yes

5. Is the manuscript presented in an intelligible fashion and written in standard English?

Reviewer #1: Yes

6. Review Comments to the Author

Reviewer #1: 1.The authors can include the following details in the discussion:Detailed Impact: Explaining in more detail how FBS-derived EVs could specifically interfere with MSC-derived EVs would strengthen the response. This might include discussing potential interactions or cross-contamination effects that could misrepresent the true properties and functions of MSC-derived EVs.

Contextual Relevance: Providing context on why purity is particularly critical in the specific research being conducted (e.g., drug delivery, diagnostic applications) would offer a more comprehensive understanding.

Comparative Analysis: Briefly mentioning what happens in the presence of EVs in FBS compared to an EV-free environment would offer a practical insight into the impact of EVs in FBS.

3.Lack of Quantification: The response, however, does not provide quantitative data on the observed decrease in TGFβ/Smad signaling. Quantification is crucial for several reasons:

Magnitude of Effect: Quantitative data would provide clarity on the extent of the decrease in TGFβ/Smad signaling. This is important for understanding the biological significance of the findings.

Comparative Analysis: It helps in comparing the extent of TGFβ/Smad signaling decrease across different methods or conditions, which could be critical for the study's conclusions.

Reproducibility and Validation: Providing exact figures or percentages allows other researchers to reproduce and validate the findings more effectively.

Potential Improvements:

Specific Data: Including specific values, such as percentages or fold-changes in TGFβ/Smad signaling levels, would significantly enhance the response.

Contextualizing the Data: Explaining how the observed changes compare to normal levels or other experimental conditions would provide a clearer understanding of the impact of UF on TGFβ/Smad signaling.

In summary, while the response addresses the importance of TGFβ/Smad signaling in MSC biology, it lacks quantification, which is crucial for assessing the significance and impact of the findings. Including specific quantitative data and contextualizing these values would greatly improve the response's informativeness and relevance.

5.Need for Quantification: The absence of specific quantitative data limits the ability to gauge the extent of the proliferation inhibition. Quantitative measures are crucial for:

Assessing Impact: Understanding the degree of inhibition (e.g., percentage reduction in cell proliferation rates compared to control or other media types).

Comparative Analysis: Comparing the effect of UF EVdFBS media on cell proliferation with other types of EV-depleted media.

Reproducibility and Validation: Enabling other researchers to reproduce the study and validate the findings.

Potential Improvements:

Specific Quantitative Measures: Including data such as percentage decrease in cell count, growth rate, or other metrics of cell proliferation would provide a clearer understanding of the 'significant inhibition'.

Statistical Significance: Adding information on the statistical significance of these findings (e.g., p-values) would strengthen the response.

Contextualizing the Findings: Providing a brief explanation of why this inhibition of cell proliferation is significant in the context of the study's goals would be beneficial. For instance, discussing how the reduced proliferation rate in UF EVdFBS media impacts the overall utility or effectiveness of this media in MSC culture and research.

7. PLOS authors have the option to publish the peer review history of their article (what does this mean?). If published, this will include your full peer review and any attached files.

Reviewer #1: No

---

## [Editor Report · Acceptance letter]

24 Nov 2023

PONE-D-23-13193R1 

Polyethylene Glycol Precipitation is an Efficient Method to Obtain Extracellular Vesicle-Depleted Fetal Bovine Serum 

Dear Dr. Wang:

I'm pleased to inform you that your manuscript has been deemed suitable for publication in PLOS ONE. Congratulations! Your manuscript is now with our production department. 

Kind regards, 

on behalf of

Dr. Wilfried A. Kues 

Academic Editor

PLOS ONE